# Self-Comparison for Dataset-Level Membership Inference in Large (Vision-)Language Model

## Abstract

Large Language Models (LLMs) and Vision-Language Models (VLMs) have made significant advancements in a wide range of natural language processing and vision-language tasks. Access to large web-scale datasets has been a key factor in their success. However, concerns have been raised about the unauthorized use of copyrighted materials and potential copyright infringement. Existing methods, such as sample-level Membership Inference Attacks (MIA) and distribution-based dataset, inference distinguish member and non-member data by leveraging the common observation that models tend to memorize and show greater confidence in member data. Nevertheless, these methods face challenges when applied to LLMs and VLMs, such as the requirement for ground-truth member data or non-member data that shares the same distribution as the test data. In this paper, we propose a novel dataset-level membership inference method based on Self-Comparison. We find that a member prefix followed by a non-member suffix (paraphrased from a member suffix) can further trigger the model's memorization on training data. Instead of directly comparing member and non-member data, we introduce paraphrasing to the second half of the sequence and evaluate how the likelihood changes before and after paraphrasing. Unlike prior approaches, our method does not require access to ground-truth member data or non-member data in identical distribution, making it more practical. Extensive experiments demonstrate that our proposed method outperforms traditional MIA and dataset inference techniques across various datasets and models, including GPT-4o.

## CCS Concepts

• **Security and privacy**;

## Keywords

Dataset protection, Membership inference, LLM, VLM

**ACM Reference Format:**
Anonymous Author(s). 2018. Self-Comparison for Dataset-Level Membership Inference in Large (Vision-)Language Model. In *Proceedings of The Web Conference 2025 (WWW '25)*. ACM, Sydney, Australia, 11 pages. https://doi.org/XXXXXXX.XXXXXXX

## 1 Introduction

Large Language Models (LLMs) [2, 43, 45] and Vision-Language Models (VLMs) [5, 26, 48] have demonstrated remarkable capabilities in understanding, reasoning, and generating both textual and visual data. These advancements have significantly impacted natural language processing (NLP) tasks such as machine translation [42], text summarization [13], and sentiment analysis [32], as well as vision-language tasks like image captioning [17], visual question answering [28], and cross-modal retrieval [24]. The rapid development of these models has been fueled by the availability of large web-scale datasets, which have substantially improved their performance. However, concerns are raised about unauthorized data usage. Copyrighted materials may be included in training datasets, potentially infringing upon the rights of content creators and causing financial loss to them [27, 36, 50]. For example, the New York Times sued OpenAI and Microsoft over the use of copyrighted work for training models[1]. These datasets often require substantial resources to construct by the companies, e.g., the New York Times, and these companies typically do not disclose them for model training purposes. Even when some datasets are open-sourced [14, 35, 38], their usage is typically restricted by licenses and is limited to educational and research purposes. Therefore, it is crucial to safeguard these datasets against unauthorized use.

To protect the datasets, it is crucial to detect their usage in training. Membership Inference Attack (MIA) [39, 40] is a widely used method in evaluating training data leakage. However, most existing research focuses on sample-level inference. When applied to dataset-level inference, MIA faces two significant challenges. *First*, current large models typically train for only one or a few epochs on extensive web-scale datasets [4, 6, 9, 15, 21, 22, 46]. As a result, each sample is encountered only a limited number of times, which restricts its contribution to the training process. Consequently, this makes inference on individual samples quite challenging [12, 29]. Existing methods demonstrate limited effectiveness in sample-level inference. *Second*, MIA often relies on a strong assumption: prior knowledge of a set of ground-truth member data. It is required either explicitly [40] or implicitly [39]. Specifically, in some methods, e.g., [39, 49], the ground-truth data is used to determine the decision threshold for the proposed MIA method, implicitly imposing the prior knowledge assumption.

To address aforementioned challenges in sample-level MIA, Maini et al. [29] propose to aggregate the membership information of individuals by using dataset distributions. They determine whether a dataset is member data based on the assumption that the distribution of MIA features, such as likelihood, for member data should significantly differ from non-member data. However, this approach is effective only when a validation set, which is non-member and has the identical distribution to the protected dataset, is available.

---

[1]https://www.nytimes.com/2023/12/27/business/media/new-york-times-open-ai-microsoft-lawsuit.html

In our preliminary study of Section 3.2, we find that even a small distribution shift between the validation data and the protected data can result in false positive detection. This highlights the necessity for improvement when there is no access to the non-member data that follows the same distribution as the protected data.

To tackle the above challenges, we propose Self-Comparison Membership Inference (SMI) for dataset-level inference. Instead of directly comparing the distributions of two sets as in [29], we focus on comparing how these distributions change under paraphrasing. Intuitively, a model should be more confident when generating member data than non-member data [10, 31, 39, 49], reflected in higher likelihood scores. For non-member data, since the model has not seen either the original or paraphrased data during training, the likelihood will remain relatively stable before and after paraphrasing. In contrast, for member data, the model has encountered the original data but not its paraphrased version, leading to a more significant likelihood change after paraphrasing. Our method does not rely on the assumption of access to ground-truth member data or non-member data that follows the same distribution as the protected data. Instead, it only requires an auxiliary non-member set which is not necessary to be in the same distribution. It can be easily obtained by synthesizing data or using data published after the release of the suspect model. In addition, our approach does not require white-box access to the model's internal parameters or architecture. It only requires access to model outputs such as logits or log probabilities, making it widely applicable even when minimal information about the model is available.

We conduct a comprehensive evaluation across various LLMs and LVMs, including publicly available model checkpoints with well-documented training data, such as Pythia [6], GPT-Neo [7], LLaVA [26], and CogVLM [48]. We also validate our approach on models that we fine-tuned. We further apply our method to verify the membership of well-known books on API-based models like GPT-4o [1]. Our extensive results demonstrate that our method outperforms existing techniques when no prior knowledge of the ground-truth member data is available.

## 2 Related works

**Sample-level MIA.** MIA is first proposed to evaluate the data leakage of individual samples by Shokri et al. [40], and is extensively studied in classification models [19, 33, 37, 47, 52]. For LLMs, recent works propose metrics based on a core assumption that the model would assign higher prediction confidence to the training data [10, 31, 39, 49, 51]. For example, Carlini et al. [10] demonstrate that member data usually has lower perplexity (greater confidence) than non-member data, and improve the detection using zlib ratio. Shi et al. [39] claim that non-member data would contain some outlier tokens with extremely low probability, and use the tokens of $k\%$ smallest likelihood to infer the membership. Mattern et al. [31] substitute synonym to evaluate the confidence in the replaced tokens. Reference-based methods [49] compare the likelihood with a reference model that is not trained on the target data. However, sample-level MIA requires ground-truth member data to assist the detection, such as training a shadow model [40] or determining a threshold [10, 31, 39, 49], which limits the practicality.

**Dataset inference.** Dataset inference is designed for a different level of membership inference. It considers from the perspective of distribution. Maini et al. [30] propose that training data is more distant from the decision boundaries in classification models. Maini et al. [29] extend the dataset inference to LLMs. They assume that the training should bring a distribution shift between member data and non-member data, and propose to use hypothesis testing for membership inference. However, their method requires ground-truth member and non-member data to train a regressor and a validation set to infer membership.

## 3 Preliminary Studies

As mentioned above, models tend to produce more confident predictions on member data [40, 50]. The more frequently a model encounters a sample during training, the more confident the prediction is. In extreme cases, this results in the memorization effect in generative models such as language models and diffusion models [10, 54]. Particularly, if a data sample is duplicated many times, the model can memorize and re-generate it [20]. Even without full memorization, training on member samples can increase a model's confidence in those samples, which can be considered a form of weak memorization [10, 29, 51].

Based on this memorization, sample-level MIA and dataset inference are proposed to detect member data. In Section 3.2, we first present the gap in the existing methods, such as distribution-based dataset inference (DDI) [29], by comparing two datasets. We show that a non-member dataset is also possible to be different from the validation set, leading to false positive detection in DDI. To further improve dataset-level membership inference, we demonstrate the possibility to fill in the gap by comparing the change of distribution after applying a paraphrase. Then in Section 3.3, we conduct an empirical study showing that a prefix sequence can further trigger the weak memorization and amplify the change of confidence.

### 3.1 Definitions

In this subsection, we define the problem and some key concepts.

**Problem definition.** Model $f$ is an auto-regressive generation model that takes a sequence of text and vision tokens as input and outputs a probability distribution predicting the next token, which is a text token. For a given dataset $Q$, the goal of dataset-level membership inference is to determine whether $Q$ was included in the training of $f$. We name $Q$ as candidate set, and $f$ as suspect model. $Q$ is composed of token sequences in varying lengths.

**Prediction confidence.** The likelihood of the predicted token can reflect the confidence of the model [10, 31, 39, 49]. For a token sequence $X = (x_1, x_2, \ldots, x_T)$, we can use average negative log-likelihood (A-NLL) to represent the confidence to generate it, which is given by

$$\text{A-NLL}(X) = -\frac{1}{T} \sum_{t=1}^{T} \log P\left(x_t \mid x_1, x_2, \ldots, x_{t-1}\right).$$

It measures how confident a model predicts a sample of text by evaluating the probability it assigns to a sequence of tokens. Lower A-NLL indicates greater confidence. In our paper, we also use a prefix sequence as input and calculate the A-NLL on the suffix, $X_{i:T} = (x_i, x_{i+1}, \ldots, x_T)$. In this case, A-NLL is only averaged on

the suffix, which is

$$\text{A-NLL}(X_{i:T}|X_{:i}) = -\frac{1}{T-i+1}\sum_{t=i}^{T}\log P\left(x_t \mid x_1, x_2, \ldots, x_{t-1}\right).$$

A-NLL is also the major term of the loss of the training of LLMs and VLMs. It can be seen as a straightforward metric to reflect the confidence and test membership. Besides A-NLL, metrics based on the intuition of confidence, such as Min-$k$% Prob [39], Max-$k$% Prob [29], perplexity [10], perturbation-based features [31], and zlib Ratio [10], are also proposed to measure membership.

## 3.2 Limitation of DDI when the validation set is not available

In this subsection, we use the DDI method proposed by [29] as an example to demonstrate the limitation when the non-member data which follows the same distribution as the protected set is not available. We begin by outlining the details of DDI [29] and then discuss the gap and the potential improvement.

The method of DDI [29] first employs a linear regressor to aggregate various sample-wise MIA metrics derived from prediction confidence, including Min-$k$% Prob [39], Max-$k$% Prob [29], perplexity [10], perturbation-based features [31], and zlib Ratio [10]. Trained on ground-truth member and non-member data, the linear regressor produces a membership score for each sample. Even though the member and non-member data are in the same distribution, the distributions of their membership scores are different. Given a candidate set $Q$, $Q_{\text{val}}$ is a validation set that is in the same distribution as $Q$ and is known to be non-member. DDI employs a $z$-test for hypothesis testing, with the null hypothesis ($H_0$) assuming $Q$ was not used in training. This $z$-test determines whether the distributions of membership scores of $Q$ and $Q_{\text{val}}$ are different. If $Q$ was not used for training, their distributions of membership scores should be similar; otherwise, they should differ. By performing a $z$-test between the membership scores of $Q$ and $Q_{\text{val}}$, if the $p$-value falls below a significance threshold (such as 0.05 or 0.01), $Q$ is classified as member data.

**Limitation of DDI.** DDI relies on a strict assumption: access to a non-member set in the same distribution as $Q$. To test whether $Q$ is a member, we must always prepare a separate set as the validation set. In practice, this is often impractical. If $Q$ is not training member, even a small difference between $Q_{\text{val}}$ and $Q$ can result in a low $p$-value, leading to a false positive prediction. Specifically, we find that the $p$-value decreases dramatically as the sizes of the two sample sets in $z$-test (such as $Q$ and $Q_{\text{val}}$ in DDI) increase, which is proved in the following Theory 1.

Theory 1. *Given two sample sets $A$ and $B$, $\mu_A$ and $\sigma_A$ are the mean and standard variance of $A$, and $\mu_B$ and $\sigma_B$ are $B$'s. Without loss of generality, we assume that the sample sizes of $A$ and $B$ are both $n$. If $\mu_A \neq \mu_B$, the $p$-value of $z$-test satisfy:*

$$p\text{-value} \propto e^{-cn},$$

*where $c$ is a constant coefficient and $c > 0$.*

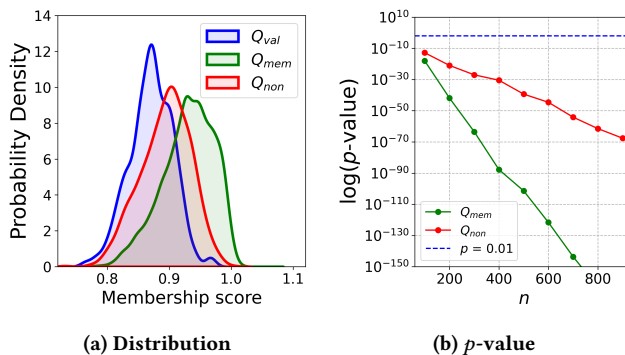

(a) Distribution      (b) $p$-value

**Figure 1: An example of using DDI**

Proof. In $z$-test, the statistic $z$ is calculated as

$$z = \frac{\mu_A - \mu_B}{\sqrt{\left(\frac{\sigma_A^2}{n}\right) + \left(\frac{\sigma_B^2}{n}\right)}} = \frac{\mu_A - \mu_B}{\sqrt{\left(\sigma_A^2\right) + \left(\sigma_B^2\right)}}\sqrt{n}. \quad (1)$$

Then the $p$-value for a one-tailed test is

$$p\text{-value } = 1 - \Phi(|z|),$$

where $\Phi(|z|)$ is the cumulative distribution function (CDF) of the normal distribution.

When $n$ increases, the tail probability $1 - \Phi(|z|)$ can be calculated using an asymptotic approximation [8, 11, 18], which gives

$$1 - \Phi(|z|) \approx \frac{1}{|z|\sqrt{2\pi}}e^{-\frac{z^2}{2}}$$

So the logarithmic of $p$-value is

$$\log(p\text{-value}) \approx \log\left(\frac{1}{|z|\sqrt{2\pi}}e^{-\frac{z^2}{2}}\right)$$

For larger $n$ (i.e. larger $|z|$), the dominant term is $-z^2$, which means $\log(p\text{-value }) \approx -\frac{z^2}{2}$. Substituting Eq. 1 into this, we have

$$\log(p\text{-value}) \propto -n, \text{ i.e., } p\text{-value} \propto e^{-cn}.$$

□

**Empirical results.** We use the example in Figure 1a to demonstrate a case of false positive detection when the distributions of $Q_{\text{val}}$ and $Q$ are not identical. Specifically, we use $Q_{\text{val}}$ to test the membership of two sets, $Q_{\text{mem}}$ and $Q_{\text{non}}$. In this scenario, $Q_{\text{mem}}$ is member data, while $Q_{\text{non}}$ is non-member data. None of them has an identical distribution to $Q_{\text{val}}$. We use Pythia-12B [6] as the suspect model, PILE [14] as $Q_{\text{mem}}$, FineWeb [34] in 2024 as $Q_{\text{non}}$ and BBC news in 2024 as $Q_{\text{val}}$. In Figure 1a, while $Q_{\text{mem}}$ differs more significantly from $Q_{\text{val}}$, there is also a small difference between $Q_{\text{non}}$ and $Q_{\text{val}}$. By calculating the $p$-value in Figure 1b, we make two key observations. *First*, the $p$-value for $Q_{\text{non}}$ is also very low, indicating a false positive detection. *Second*, the logarithm of the $p$-value decreases linearly as the sample size $n$ increases, which aligns with Theory 1. The difference between $\overline{Q}_{\text{val}}$ and $\overline{Q}_{\text{non}}$ may arise from various factors, such as the lack of non-member data that follows the same distribution and the randomness of the sampling process.

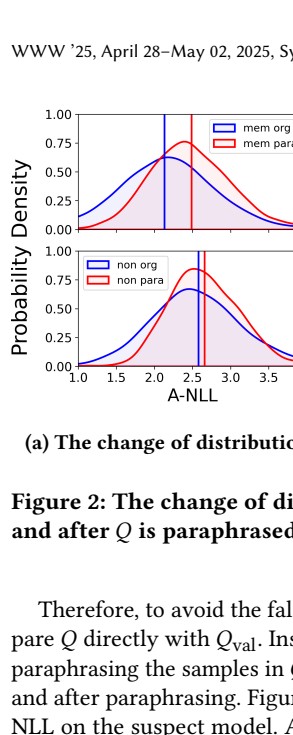
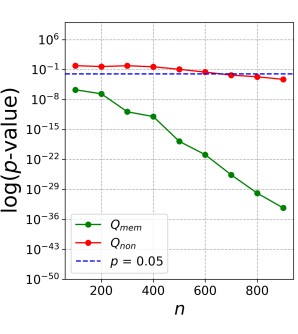

(a) The change of distribution

(b) $p$-value

Figure 2: The change of distribution and $p$-values of before and after $Q$ is paraphrased.

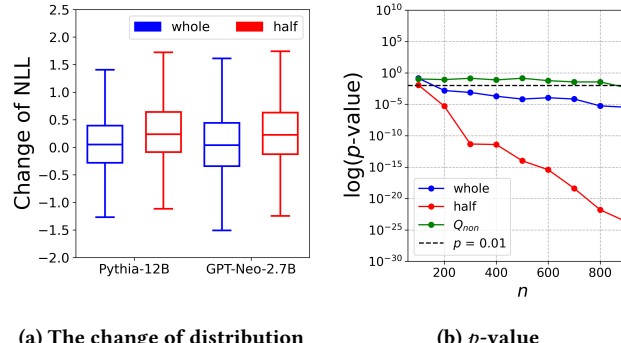

(a) The change of distribution

(b) $p$-value

Figure 3: Different memorization results between paraphrasing the whole sequence and half of the sequence.

Therefore, to avoid the false positive detection, we do not compare $Q$ directly with $Q_{\mathrm{val}}$. Instead, we introduce changes through paraphrasing the samples in $Q$ and compare the differences before and after paraphrasing. Figure 2a illustrates the distribution of A-NLL on the suspect model. After paraphrasing, the mean A-NLL of $Q_{\mathrm{mem}}$ increases more significantly than that of $Q_{\mathrm{non}}$. This is because the model has encountered the member data during training, resulting in (weak) memorization. After paraphrasing, the verbatim member data becomes non-member data, leading to an increase in A-NLL. In contrast, non-member data is less affected by paraphrasing, as the model has never been trained on it. In Figure 2b, we also conduct a $z$-test to compare the distributions before and after paraphrasing. Denote $\mu_{\mathrm{org}}$ as the mean of A-NLL of the sample set *before* paraphrase and $\mu_{\mathrm{para}}$ as the mean of A-NLL of the sample set *after* paraphrase. Then, the null hypothesis and alternative hypothesis ($H_1$) can be formulated as

$$H_0 : \mu_{\mathrm{org}} \geq \mu_{\mathrm{para}}; \quad H_1 : \mu_{\mathrm{org}} < \mu_{\mathrm{para}}$$

Lower $p$-value indicates a more significant change after paraphrasing. In Figure 2b, the $p$-value for $Q_{\mathrm{mem}}$ decreases much more rapidly, demonstrating that $Q_{\mathrm{mem}}$ undergoes greater change after paraphrasing compared to $Q_{\mathrm{non}}$. In the following subsection, we further provide a study on when will the change become more obvious.

## 3.3 Prefix Sequences can trigger memorization on training Data

As we analyzed, the model exhibits greater confidence (i.e., lower A-NLL) on data it has encountered during training. However, in this subsection, we observe that paraphrasing the entire sequence of member data does not always result in a high A-NLL, making it difficult to distinguish from non-member data. In this subsection, we provide a detailed analysis of this phenomenon and point out how to exemplify the change of A-NLL caused by the paraphrase introduced in the previous subsection.

In Figure 3a, we present the change in A-NLL after applying two different paraphrasing methods: whole paraphrase and half paraphrase. For the whole paraphrase (blue in Figure 3a), we paraphrase the whole text sequences. For each sample, we calculate the NLLs of both the original and paraphrased text, then subtract the original NLL from the paraphrased NLL and plot the results.

For the half paraphrase (red in Figure 3a), we leave the first half of the text unchanged, paraphrasing only the second half, and report the change in NLL for the paraphrased portion. As shown, for the whole paraphrase, the average change in NLL is slightly above 0, indicating that paraphrasing causes only a minor increase in NLL. In contrast, the half paraphrase leads to a much more significant increase in NLL. Consequently, in Figure 3b, the $p$-value for the half paraphrase decreases more rapidly, making it easier to distinguish from non-member data.

We conjecture that half paraphrasing can introduce an abrupt and unexpected change in the prediction sequence. When given a prefix, the causal model relies on it to predict the likelihood distribution of the next token. If the prefix comes from member data, the model is likely to follow its memorization verbatim, even if that memorization is weak. Paraphrasing the second half can disrupt this expectation, leading to a higher NLL. Conversely, if the prefix is from non-member data, the paraphrased text still appears reasonable, and the model does not have a strong tendency to predict the original text verbatim. In contrast, the whole paraphrasing does not induce such unexpected changes, as no member prefix exists to trigger a verbatim prediction based on memorization.

For VLMs, the image tokens in the input can be viewed as the prefix. We will present the difference of the processing between LLMs and VLMs in the following section.

## 4 Method

Based on the observations from the preliminary studies, in this section, we propose our method, Self-Comparison Membership Inference (SMI). We first introduce the framework of Self-Comparison to get the $p$-values in Section 4.1. Then we explain how to use the $p$-values and its trend as $n$ changes to determine membership in Section 4.2. Lastly, we introduce a variant for a challenging case where only the probability of the predicted token is available in Section 4.3.

### 4.1 Self-Comparison

Building on preliminary studies, we propose SMI to infer the membership by analyzing the change of A-NLL distribution. To caption this distribution shift, SMI uses the $p$-value derived from the hypothesis testing between the datasets before and after paraphrasing.

**Figure 4: The framework of Self-Comparison**

In this subsection, we provide a detailed explanation of the hypothesis testing process and how the $p$-values are calculated. We name the process to get the $p$-values as Self-Comparison. The framework of Self-Comparison can be found in Figure 4, which is performed in three key steps. We begin by detailing these steps using textual datasets, specifically focusing on membership inference in LLMs. Following this, we discuss how the framework adapts to multi-modal datasets, such as VQA and captioning, to demonstrate the framework for VLMs.

**LLMs.** As discussed in Section 3.3, to make the change of member data more obvious and easier to capture, SMI takes advantage of triggered memorization of prefix sequences. For a given set $Q$, Self-Comparison processes it using the following steps.

(1) *Half paraphrase.* We first truncate all the samples of $Q$ into sequences with a length of 150 tokens, and then paraphrase the second half of the sequence. To ensure the completeness of sentences, we remove the last sentence if it is broken resulting from truncation. To get a complete second half, we segment the sequence by sentences rather than tokens. We use Gemma 2 [44] for paraphrase. The paraphrase prompt can be found in Appendix A.1.

(2) *Membership Metric calculation.* In the proposed SMI, we use the most straightforward A-NLL as the metric. We calculate the average NLL on the tokens in the second half of the sequence. By inputting the original data and paraphrased data into the LLMs, we obtain two sets of A-NLL values. (It is worth noting that there is a challenge with certain models, such as ChatGPT-4, which only return the log probability for the predicted token. We provide a comprehensive discussion of this case in Section 4.3.)

(3) *Comparison.* Based on the preliminary studies, we know that if the model is trained on $Q$, the distribution of A-NLL should have a significant change. We calculate the $p$-values of the hypothesis testing with $H_0$ that $Q$ is not member data. To get the trend of $p$-value as the sample size increases, we calculate a series of $K$ $p$-values, $\{p_i \mid 1 \le i \le K\}$ at $K$ equal intervals, $\{n_i \mid 1 \le i \le K\}$, where $n_i = \frac{i}{K}N$ and $N$ is the total number of samples in $Q$. We use the slope of linear least-squares regression for $p_i$ and $n_i$ (such as the orange dash line in $p$-value of $Q_{\text{mem}}$ in Figure 4) to represent the trend of $p$-values. The slope can be denoted

as $\beta = \dfrac{\sum_{i=1}^{K}(n_i - \overline{n})\left(\log p_i - \overline{\log p}\right)}{\sum_{i=1}^{K}(n_i - \overline{n})^2}$, where $\overline{n}$ is the mean of $\{n_i \mid 1 \le i \le K\}$ and $\overline{\log p}$ is the mean of $\{\log p_i \mid 1 \le i \le K\}$.

It is worth mentioning that, based on our experiments in Section 5, in SMI, $N = 500$ is large enough to provide a solid inference. This means that for web-scale large datasets, we do not need to test the whole dataset. We can sample a set of size $N$, which is efficient and effective to verify the unauthorized dataset usage.

**VLMs.** For datasets such as VQA and image captioning, image is also a part of input. This leads to a small difference in the step of paraphrase. For such datasets, SMI keeps the image tokens and questions unchanged and paraphrases the textual response. For the datasets with multi-round chatting, we only use the first round. For step (2) and (3), it is the same as LLMs.

After obtaining the results of $p$-values, in the following subsection, we present the determination conditions for classifying $Q$ as a training member.

## 4.2 Criteria for membership

Although we use hypothesis testing to get $p$-values, the traditional significance level like $p = 0.01$ is not applicable to our problem. As we mentioned in Section 3.2, a small difference may lead to a small $p$-value of $Q_{\text{non}}$ especially when $n$ is large. The paraphrase may also bring in such small differences. In this subsection, for a more rigorous membership inference, we propose to use an auxiliary dataset $Q_{\text{aux}}$ to eliminate the impact of paraphrase and present the two criteria for membership using $Q_{\text{aux}}$.

SMI uses a non-member set as the auxiliary set, $Q_{\text{aux}}$. This $Q_{\text{aux}}$ is non-member, but not necessary to be the same distribution to $Q$. It is easy to obtain from synthesized data, unpublished data, and the data released after the model training date. We denote the $p$-values from Self-Comparison of $Q_{\text{aux}}$ as $\{p'_i \mid 1 \le i \le K\}$, and its slope as $\beta'$. In the following experiments of Section 5, we observe that $p'_i$ is usually larger than or close to 0.01 and $\beta'$ is close to 0. Based on $Q_{\text{aux}}$, we say that $Q$ is the member data in the training set when it satisfies the following two criteria:

$$\beta < \beta' - \epsilon_1, \tag{2}$$

$$\log p_K < \log p'_K - \epsilon_2. \tag{3}$$

Both of the criteria indicate the change of A-NLL of $Q$ should be more significant than $Q_{\text{aux}}$. Eq. 2 means the slope of $Q$ should be smaller than $Q_{\text{aux}}$ by $\epsilon_1$, while Eq. 3 means the $p$-value of $Q$ should

be smaller than $Q_{aux}$ by $\epsilon_2$. The constants, $\epsilon_1$ and $\epsilon_2$, are two margin values to reduce randomness. They are not data-specific since the $z$-test is not data-specific for various datasets. This is different from the threshold in sample-wise MIA methods.

**Remarks.** From the above pipeline of Self-Comparison and the criteria, we can find that SMI does not rely on ground-truth member data. This relaxes the strict assumptions and makes the dataset-level inference much more practical.

### 4.3 A variant when not the logits/probability of the whole vocabulary are available

For some models such as the API of Together AI platform[2], the logits for the whole vocabulary at each token position are available. For example, if the previous sequence is "*Today is a sunny ...*" and the next token is predicted to be "*day*", these models will provide not only the probability of "*day*", but also the probability of the whole vocabulary. However, for other models like GPT-4o, they only provide the probability of the predicted token. For example, if $Q$ has a sample that is "*Today is a sunny and warm day*", we need the probability of "*and warm day*" to calculate A-NLL. But the model prediction is "*Today is a sunny day*". It only provides the probability of "*day*", but we need the probability of "*and*". This is a more difficult scenario.

To solve this challenge, we propose to use a constant as the probability of the unavailable tokens. To get the probability of "*and warm day*", we first input the prefix "*Today is a sunny*", if the model prediction on the next token is "*and*", we can have the probability of "*and*". However, if the prediction is "*day*", it means the probability of "*and*" should be low. Thus, we assign a low constant probability value (i.e., large NLL) to it. Then we add "*and*" into the prefix and continue to input "*Today is a sunny and*" to predict "*warm*" until we iterate through all the following tokens. With this improvement, we can also use SMI when only the probability of the predicted token is available.

## 5 Experiments

In this section, we present the experiments to show the effectiveness of SMI. We conduct the experiments across different models and datasets in Section 5.2, and ablation studies including model size, sample size, and margin values in Section 5.3. Additionally, in Appendix B, we also conduct the experiment to demonstrate the effectiveness of SMI when only part of $Q$ is used for training. Next, we first introduce the experimental settings.

### 5.1 Experimental Settings

**Suspect models and datasets.** We conduct experiments on public model checkpoints, models fine-tuned by ourselves, and API-based commercial models. The details are as follows:

*(1)* For public checkpoints, we include **two LLMs**, Pythia [6] and GPT-Neo [7], and **two VLMs**, LLaVA [26] and CogVLM [48]. The training dataset of Pythia and GPT-Neo is PILE [14], and we use New Yorker Caption Contest (NY) [16] and FineWeb (F-CC) [34] crawled in 2024 as $Q_{non}$ and $Q_{aux}$. The training data of LLaVA includes TextVQA (TVQA) [41], Visual Genome (VG) [23], and MS COCO

[2]https://www.together.ai/

(COCO) [25]. The training data of CogVLM includes CogVLM-SFT (Cog) [48]. And we use NoCaps (NC) [3] and Flickr (Flkr) [53] as $Q_{non}$ and $Q_{aux}$ for public VLMs.

*(2)* For fine-tuned models, we train Pythia-1.4B on FineWeb, NY, or BBC news in 2024. We train LLaVA using Flickr or MS COCO. To simulate the real-world fine-tuning, we add 118,000 unrelated samples into the training set of Pythia-1.4B and 20,000 into LLaVA. Each model is trained in one epoch.

*(3)* For API-based models, we use GPT-4o. Since $Q_{mem}$ is unknown, we use famous books, *Bible*, *Pride and Prejudice* (Pride), and *Harry Potter* (HP), and BookMIA containing member (B-Pos) and non-member (B-Neg) samples of OpenAI models. The non member in BookMIA (B-Neg) and BBC news in 2024 are $Q_{non}$ and $Q_{aux}$.

To further validate SMI, we use different datasets as $Q_{non}$ and $Q_{aux}$ alternatively.

**Baselines.** To ensure the fairness of experiments, we assume all the baselines and SMI have no access to the ground-truth member data, but have access to $Q_{aux}$ which is non-member and does not necessarily follow the same distribution as member data. The baseline methods include DDI and three sample-level MIAs, A-NLL, Min-$k$% [39] and zlib ratio [10]. Each sample-level MIA calculates a membership score and uses it to determine the membership of one sample. We use the 45th percentile of the membership score of $Q_{aux}$ as the threshold (which is explained in Figure 5 and Section 5.2). In the three MIAs, lower membership scores than the threshold will be classified as member. In addition, to better match the sample-level MIAs to the dataset-level tasks, we create variants of them by counting the positives in $Q$. We first use sample-level MIAs to classify each sample in $Q$. If more than 50% samples are predicted as member by the sample-wise MIA, we classify $Q$ as member. Otherwise, it is classified as non-member.

**Evaluation metrics.** We use F1 score, recall, and precision as evaluation metrics. In each original dataset, there are at least 900 samples. For dataset-level inference, to increase the number of datasets and get more convincing results, we construct 300 sub-sets from the original datasets, with each sub-set consisting of 500 randomly sampled sequences ($N = 500$) from the original dataset. The 300 sub-sets are composed of 100 $Q_{mem}$s, 100 $Q_{non}$s and 100 $Q_{aux}$s. The metrics are calculated by the label and prediction for the sub-sets. In addition, to better explain the variants of sample-level MIA, we also calculate the evaluation metrics at the sample level using all the samples in the original datasets and use them as a reference.

**Implementation details.** For all the models, we only use the output probability for membership inference. For all the results of SMI, we use $\epsilon_1 = 0.01$ and $\epsilon_2 = 10$. For GPT-4o, we use the chat template in Appendix A.2.

### 5.2 Main results

In this subsection, we show the effectiveness of our method in different LLMs and VLMs. We reported the results of public model checkpoints, the model fine-tuned by ourselves, and API-based GPT-4o in Table 1. We get two observations from the results.

**1. SMI outperforms all the baseline methods across various models and datasets.** For *public models and fine-tuned models*, Table 1 shows that the average F1 scores of SMI consistently exceed

**Table 1: Dataset-level membership inference on public and fine-tuned models. The results are F1 score (recall/precision). We label the best average F1 score by bold fonts.**

| Public LLM $Q_{mem}/Q_{non}/Q_{aux}$ | Pythia-1.4B PILE/F-CC/NY | Pythia-6.9B PILE/NY/F-CC | Pythia-12B PILE/F-CC/NY | GPT-Neo-1.3B PILE/NY/F-CC | GPT-Neo-2.7B PILE/F-CC/NY | Average |
|---|---|---|---|---|---|---|
| A-NLL (Dataset) | 0.667 (1.000/0.500) | 1.000 (1.000/1.000) | 0.667 (1.000/0.500) | 1.000 (1.000/1.000) | 0.667 (1.000/0.500) | 0.800 (1.000/0.700) |
| Min-$k$% (Dataset) | 0.667 (1.000/0.500) | 1.000 (1.000/1.000) | 0.667 (1.000/0.500) | 1.000 (1.000/1.000) | 0.667 (1.000/0.500) | 0.800 (1.000/0.700) |
| zlib (Dataset) | 0.667 (1.000/0.500) | 1.000 (1.000/1.000) | 0.667 (1.000/0.500) | 1.000 (1.000/1.000) | 0.667 (1.000/0.500) | 0.800 (1.000/0.700) |
| DDI | 0.667 (1.000/0.500) | 0.667 (1.000/0.500) | 0.667 (1.000/0.500) | 0.667 (1.000/0.500) | 0.667 (1.000/0.500) | 0.667 (1.000/0.500) |
| SMI (ours) | 0.958 (0.920/1.000) | 1.000 (1.000/1.000) | 0.995 (1.000/0.990) | 0.980 (0.970/0.990) | 0.985 (0.970/1.000) | **0.984** (0.972/0.996) |

| Public VLM $Q_{mem}/Q_{non}/Q_{aux}$ TVQA/Flkr/NC | LLaVA-v1.5 VG/NC/Flkr | COCO/Flkr/NC | CogVLM-v1 Cog/NC/Flkr | CogVLM-v1-chat Cog/Flkr/NC | Average |
|---|---|---|---|---|---|
| A-NLL (Dataset) 0.667 (1.000/0.500) | 1.000 (1.000/1.000) | 0.669 (1.000/0.503) | 0.000 (0.000/0.000) | 0.667 (1.000/0.500) | 0.600 (0.800/0.501) |
| Min-$k$% (Dataset) 1.000 (1.000/1.000) | 1.000 (1.000/1.000) | 0.995 (1.000/0.990) | 0.000 (0.000/0.000) | 0.667 (1.000/0.500) | 0.732 (0.800/0.698) |
| zlib (Dataset) 0.667 (1.000/0.500) | 1.000 (1.000/1.000) | 0.667 (1.000/0.500) | 1.000 (1.000/1.000) | 0.667 (1.000/0.500) | 0.800 (1.000/0.700) |
| DDI 0.667 (1.000/0.500) | 0.873 (1.000/0.775) | 1.000 (1.000/1.000) | 0.667 (1.000/0.500) | 0.667 (1.000/0.500) | 0.775 (1.000/0.655) |
| SMI (ours) 0.980 (0.990/0.971) | 1.000 (1.000/1.000) | 0.980 (1.000/0.962) | 1.000 (1.000/1.000) | 1.000 (1.000/1.000) | **0.992** (0.998/0.986) |

| Fine-tuned $Q_{mem}/Q_{non}/Q_{aux}$ F-CC/NY/BBC | Pythia-1.4B NY/BBC/F-CC | BBC/F-CC/NY | LLaVA (initialized by Vicuna) Flkr/NC/TC | COCO/Flkr/VG | Average |
|---|---|---|---|---|---|
| A-NLL (Dataset) 0.667 (1.000/0.500) | 1.000 (1.000/1.000) | 1.000 (1.000/1.000) | 0.667 (1.000/0.500) | 1.000 (1.000/1.000) | 0.867 (1.000/0.800) |
| Min-$k$% (Dataset) 0.667 (1.000/0.500) | 1.000 (1.000/1.000) | 1.000 (1.000/1.000) | 0.667 (1.000/0.500) | 1.000 (1.000/1.000) | 0.867 (1.000/0.800) |
| zlib (Dataset) 0.667 (1.000/0.500) | 1.000 (1.000/1.000) | 1.000 (1.000/1.000) | 0.667 (1.000/0.500) | 0.667 (1.000/0.500) | 0.800 (1.000/0.700) |
| DDI 0.667 (1.000/0.500) | 0.694 (1.000/0.532) | 0.667 (1.000/0.500) | 0.667 (1.000/0.500) | 0.667 (1.000/0.500) | 0.672 (1.000/0.506) |
| SMI (ours) 1.000 (1.000/1.000) | 0.995 (1.000/0.990) | 1.000 (1.000/1.000) | 0.971 (1.000/0.943) | 1.000 (1.000/1.000) | **0.993** (1.000/0.987) |

| API-based $Q_{mem}/Q_{non}/Q_{aux}$ Bible/B-Neg/BBC | GPT-4o Pride/B-Neg/BBC | HP/B-Neg/BBC | B-Pos/B-Neg/BBC | Average |
|---|---|---|---|---|
| A-NLL (Dataset) 1.000 (1.000/1.000) | 0.000 (0.000/0.000) | 0.000 (0.000/0.000) | 0.000 (0.000/0.000) | 0.250 (0.250/0.250) |
| Min-$k$% (Dataset) 1.000 (1.000/1.000) | 0.000 (0.000/0.000) | 0.000 (0.000/0.000) | 0.000 (0.000/0.000) | 0.250 (0.250/0.250) |
| zlib (Dataset) 0.000 (0.000/0.000) | 0.667 (1.000/0.500) | 0.667 (1.000/0.500) | 0.667 (1.000/0.500) | 0.500 (0.750/0.375) |
| DDI 0.667 (1.000/0.500) | 0.667 (1.000/0.500) | 0.667 (1.000/0.500) | 0.667 (1.000/0.500) | 0.667 (1.000/0.500) |
| SMI (ours) 1.000 (1.000/1.000) | 1.000 (1.000/1.000) | 0.919 (0.850/1.000) | 0.958 (0.920/1.000) | **0.969** (0.943/1.000) |

**Table 2: Sample-level MIAs. The results are F1 score (recall/precision).**

| Public LLM $Q_{mem}/Q_{non}/Q_{aux}$ | Pythia-1.4B PILE/F-CC/NY | Pythia-6.9B PILE/NY/F-CC | Pythia-12B PILE/F-CC/NY | GPT-Neo-1.3B PILE/NY/F-CC | GPT-Neo-2.7B PILE/F-CC/NY | Average |
|---|---|---|---|---|---|---|
| A-NLL | 0.684 (0.877/0.561) | 0.785 (0.690/0.910) | 0.689 (0.898/0.559) | 0.783 (0.698/0.893) | 0.682 (0.882/0.556) | 0.725 (0.809/0.696) |
| Min-$k$% | 0.645 (0.756/0.563) | 0.720 (0.658/0.794) | 0.666 (0.803/0.569) | 0.702 (0.649/0.764) | 0.661 (0.790/0.568) | 0.679 (0.731/0.651) |
| zlib | 0.670 (0.886/0.539) | 0.778 (0.659/0.951) | 0.677 (0.906/0.541) | 0.782 (0.659/0.961) | 0.673 (0.892/0.540) | 0.716 (0.800/0.706) |

| Public VLM $Q_{mem}/Q_{non}/Q_{aux}$ TVQA/Flkr/NC | LLaVA-v1.5 VG/NC/Flkr | COCO/Flkr/NC | CogVLM-v1 Cog/NC/Flkr | CogVLM-v1-chat Cog/Flkr/NC | Average |
|---|---|---|---|---|---|
| A-NLL 0.701 (0.830/0.606) | 0.822 (0.953/0.723) | 0.780 (0.983/0.646) | 0.371 (0.320/0.440) | 0.737 (0.992/0.586) | 0.682 (0.816/0.600) |
| Min-$k$% 0.701 (0.787/0.632) | 0.799 (0.961/0.683) | 0.795 (0.962/0.677) | 0.299 (0.251/0.371) | 0.712 (0.881/0.598) | 0.661 (0.768/0.592) |
| zlib 0.671 (0.865/0.549) | 0.765 (0.755/0.776) | 0.729 (0.982/0.580) | 0.861 (0.935/0.797) | 0.720 (0.999/0.563) | 0.749 (0.907/0.653) |

0.98, demonstrating that our method can accurately distinguish member data. In contrast, while dataset-level MIA variants achieve high F1 scores on some models, such as Pythia-6.9B and GPT-Neo-1.3B, they perform poorly on others like Pythia-1.4B and CogVLM-v1, resulting in significantly lower F1 scores and precision compared to our method. This is because previous MIA approaches rely on

data-specific thresholds that are not consistently applicable across different models and datasets (which are detailed below). As for DDI, its performance is typically 0.667 (1.000/0.500), as it classifies every $Q$ as member data regardless of its true label. While this allows it to recall all member sets, it misclassifies all non-member sets as member, leading to a precision of 0.5. For *API-based GPT-4o*, our

 

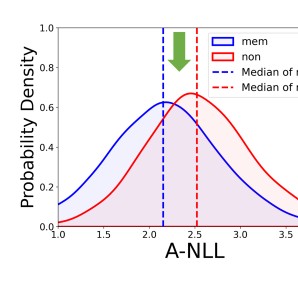

Figure 5: Three regions for the threshold of sample-level MIAs

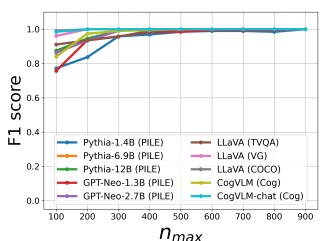

Figure 6: F1 score of SMI on different sample sizes

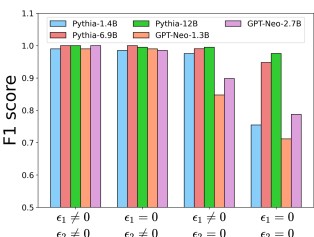

Figure 7: F1 score of SMI on different margin values

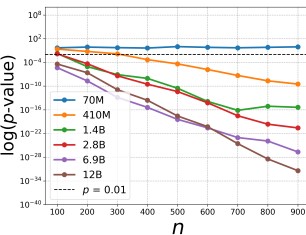

Figure 8: $p$-values of SMI on different model sizes

method significantly outperforms others, achieving an F1 score of 0.969. The performance of dataset-level MIA variants is worse than their performance on public and fine-tuned models. This difference is likely due to the lack of access to the probabilities of all tokens, which may affect the accuracy of their membership scores.

**2. The performance of sample-level MIAs and their dataset-level variants is highly dependent on the choice of threshold.** From Table 1, it is evident that the dataset-level variants of MIAs, such as A-NLL (Dataset), often exhibit three distinct modes of F1 scores: 0.000, 0.667, and 1.000. These modes are a result of three different threshold choices. In Figure 5, we plot the distributions of A-NLL of $Q_{\mathrm{mem}}$ and $Q_{\mathrm{non}}$ and their medians. We can see that the two medians split the x-axis into three regions. When the threshold falls within the middle region (between the medians), indicated by the green arrow in Figure 5, more than 50% member samples will be correctly classified as member. Since all the samples in $Q_{\mathrm{mem}}$ are member samples, $Q_{\mathrm{mem}}$ will be classified as member. Similarly, less than 50% non-member data will be classified as member and $Q_{\mathrm{non}}$ will be classified as non-member. This leads to the correct classification of every $Q$, and F1 score is 1.000. However, if the threshold falls in the right region, all the $Q$s will be classified as member set. In this case, the F1 score drops to 0.667 (1.000/0.500), which is similar to DDI, i.e., all the member sets are recalled, but the precision is only 0.5. On the other hand, if the threshold is in the left region, all $Q$s are classified as non-member sets, meaning no member sets are recalled, leading to an F1 score of 0.000 (0.000/0.000). To choose the threshold from the middle region, we conjecture that the distribution of $Q_{\mathrm{aux}}$ is more similar to $Q_{\mathrm{non}}$ since they are both non-member. Then the middle region is possible to locate at the left of the median of $Q_{\mathrm{aux}}$. Thus, we use the 45th percentile of the membership score of $Q_{\mathrm{aux}}$ as the threshold.

In addition, it is important to note that there is a significant overlap between the distributions of $Q_{\mathrm{mem}}$ and $Q_{\mathrm{non}}$, which means it is hard to use a threshold to distinguish the member and non-member samples. We conduct sample-level inference experiments using sample-level MIAs and present the results of sample-level MIAs in Table 2. As shown, most of the sample-level MIAs achieve F1 scores of 0.65 to 0.7 on public models, (which is very low since random guess has F1 score of 0.5, and predicting all the samples as member is 0.667). The observed overlap helps to explain the poor performance of existing sample-level MIAs.

In summary, our method can achieve significantly better performance than baseline methods and does not rely on ground-truth data to determine a data-specific threshold.

## 5.3 Ablation studies

In this subsection, we conduct ablation studies on sample size, margin values and model size.

**Sample size.** In some scenarios, models may restrict the number of allowed queries. Verifying membership with fewer samples becomes important. In Figure 6, we show F1 scores of SMI on public models when fewer samples are available. Recall that we use $N$ to represent the number of the total available samples. The results demonstrate that the performance is stable and high when $N \geq 300$. Even with $N \geq 100$, SMI achieves F1 scores above 0.76 across all models, with most scores around 0.9. In summary, our method maintains strong performance even with a limited sample size.

**Margin values.** In Figure 7, we demonstrate the effectiveness of the two margin values, $\epsilon_1$ and $\epsilon_2$. The results show that both $\epsilon_1$ and $\epsilon_2$ improve performance compared to the case without margin values. Using $\epsilon_2$ alone yields a higher F1 score than using $\epsilon_1$ alone. Furthermore, combining both margin values leads to an even better performance in the F1 score. This improvement occurs because the margin values help reduce noise in the sampling process.

**Model size.** Larger models usually have more parameters which might memorize larger datasets. In Figure 8, we present the trend of $p$-values for the Pythia series across different model sizes. The results indicate that as model size increases, the $p$-value decreases faster. Notably, this decrease is evident even at a model size of 410M, demonstrating that our method can effectively distinguish member data at this scale. In contrast, smaller models, such as Pythia-70M, exhibit insufficient capacity to memorize large datasets. These smaller models do not raise significant concerns as they cannot effectively leverage the knowledge and corpus of the dataset.

## 6 Conclusion

In this paper, we propose a novel dataset-level membership inference method based on Self-Comparison. Instead of directly comparing member and non-member data, our approach leverages paraphrasing on the second half of the sequence and evaluates how the likelihood changes before and after paraphrasing. Unlike prior approaches, our method does not require access to ground-truth member data or non-member data in identical distribution, enhancing its practicality. Extensive experiments demonstrate that our proposed method outperforms traditional MIA and dataset inference techniques across various datasets and models, including public models, fine-tuned models, and API-based commercial models.

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

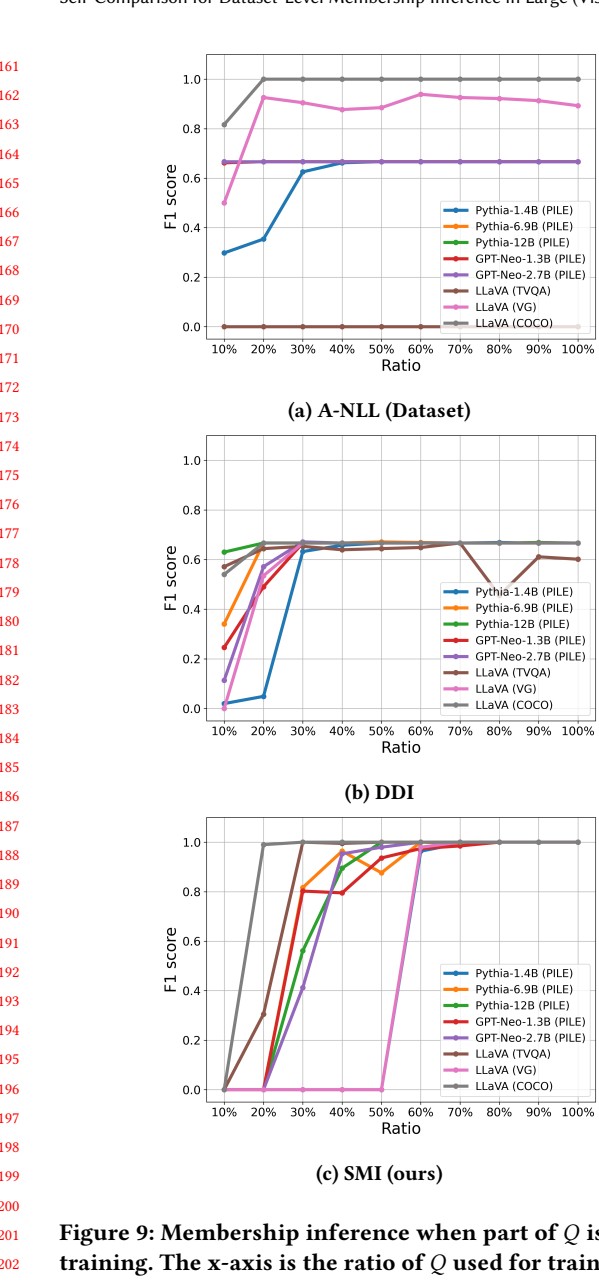

(a) A-NLL (Dataset)

(b) DDI

(c) SMI (ours)

**Figure 9: Membership inference when part of $Q$ is used for training. The x-axis is the ratio of $Q$ used for training.**

# A  Details on the prompts used in SMI

## A.1  Paraphrasing

The prompt of paraphrasing using Gemma 2 is:

"*You need to paraphrase the sentences that user gives. Directly output the paraphrased texts! The input for you to paraphrase is as follows:*"

## A.2  GPT-4o

The prompt for GPT-4o is :

"*Please complete the following sentence. Output the next words directly! The incomplete sentence is:*"

# B  Additional experiments

We conduct experiments to test the membership inference performance when part of data in $Q$ is used for training, and plot the results in Figure 9. To simulate the training with partial $Q$, we do not directly re-train a model with part of $Q$. Instead, we use the public model checkpoints, but mix non-member data into $Q$ in the membership inference process. For instance, if $r\%$ of $Q$ consists of non-member data, the result can be interpreted as the suspect model using only $r\%$ of $Q$ for training.

From the results, we can see that when more than 40% is used for training, on most of models, our method can perform well with F1 score higher than 0.9. In contrast, A-NLL (Dataset) is still suffering from the problem of threshold at all ratios. Without the prior knowledge to the ground-truth member data or data-specific threshold, the performance is inconsistent on different models. As for DDI, the membership inference still has a high false positive rate. In summary, our method still outperforms baseline methods when only part of $Q$ is used for training.

Received 20 February 2007; revised 12 March 2009; accepted 5 June 2009

