# OpenReview forum: "Self-Comparison for Dataset-Level Membership Inference in Large (Vision-)Language Model"
_ACM.org/TheWebConf/2025/Conference — WWW 2025 Oral_

### Official Review · Reviewer_4znB · 2024-11-22

**Novelty:** 4
**Technical Quality:** 3

**Review:**

This paper studies Membership Inference Attack (MIA) for evaluating training data leakage in LLMs.

Pros:
1. This paper proposes systematically analyzes the limitations of existing methods.
2. This paper proposes simple yet effective method for MIA, especially for open-sourced LLMs.
3. The reported experimental results shows significantly improvement in accuracy for MIA of the proposed method, compared to baselines.

Cons:
1. Closed-sourced models can refuse to provide the logits/probabilities of the whole vocabulary to defend the proposed method.
2. Closed-sourced models can refuse to provide the vocabulary/tokenizer to defend the proposed method.
3. The experiments only includes relatively small open-sourced LLMs. The sensitivity of the method to the size of LLMs is unclear.
4. Case study of sample-level MIA should be provided..

**Questions:**

Please respond to my listed cons.

**Reviewer Confidence:**

3: The reviewer is confident but not certain that the evaluation is correct

**Scope:**

3: The work is somewhat relevant to the Web and to the track, and is of narrow interest to a sub-community

---

### Official Review · Reviewer_UUh7 · 2024-11-26

**Novelty:** 5
**Technical Quality:** 5

**Review:**

**Summary:**
This paper proposes a method for Dataset-level Membership Inference in a Large Language Model (LLM) by introducing a novel mechanism called Self-Comparison. By comparing the differences in distribution shift after paraphrasing the validation dataset, Self-Comparision Membership Inference (SMI) can tell if the dataset is learned by this model inspired by empirical analysis. SMI achieves the highest F1 score (0.958~1.0) with four state-of-the-art (SOTA) membership inference methods across four LLM models with five different datasets.

**Strengths:**

++  SMI is a simple and efficient membership inference method

++ Unlike previous works, SMI does not assume the validation dataset shares the same distribution as the training dataset.

++ Extensive experiments have justified the effectiveness of SMI.

**Questions:**

**Weaknesses:**

-- Lack of formal justification for the assumption that paraphrasing can make the distribution shift more evident for non-member data.

Please provide proofs like Theory 1 or solid justification for this assumption, since only empirical analysis is not convincing enough.

-- Lack of details of experiment settings

In section 3.3 the green curve in Figure~2(b) is evidently different than that in Figure~2(b). Please specify the setting in this section to avoid confusion.

In lines 673--677, please add a reference to related works that apply the same setting for fair comparisons.

-- Lack of convincing examples

In lines 117--119, the author states `In our preliminary study of Section 3.2, we find that even a small distribution shift between the validation data and the protected data can result in false positive detection'. However, the example provided for the membership inference attack does not represent this problem as a realistic challenge. Please provide an alternative example to better justify this as a practical and realistic challenge.

-- The relationship between dataset inference and sample-level MIA is not formally discussed.

The relationship between dataset inference and sample-level MIA is obfuscated without a formal definition, and further discussion is not provided here.

Please add more details on the relationship between sample-level MIA and the dataset inference to avoid confusion.

**Detail Comments**

See comments above.

**Writing Issues**

-- In lines 520--521, please explain the purpose of deriving the trend of p-values in your proposed SMI. It's recommended to add it to your pipeline cartoon (i.e. Figure~4). This step looks confusing to me.

-- Do the same for Section 4.2.

**Reviewer Confidence:**

3: The reviewer is confident but not certain that the evaluation is correct

**Scope:**

3: The work is somewhat relevant to the Web and to the track, and is of narrow interest to a sub-community

---

### Official Review · Reviewer_1S2g · 2024-11-28

**Novelty:** 4
**Technical Quality:** 4

**Review:**

**Review**

This paper introduces SMI, a method for determining whether a specific dataset was used in training LLMs and VLMs, addressing the limitations of existing techniques by analyzing changes in prediction confidence after paraphrasing portions of the input data.

**Pros**:

1. SMI only requires access to model outputs such as logits or log probabilities, making it applicable even when minimal information about the model is available.

2. Conducted a comprehensive evaluation across various LLMs and LVMs.
3. The introduction to the preliminary studies is detailed and easy to understand.

**Cons:**

1. The paper lacks an explanation of how the topic is related to the web, which would align it better with the conference theme.

2. The information presented in the framework diagram (Fig. 4) is insufficient, and its size is disproportionate to the amount of content, leading to sparse information and an unappealing presentation.

3. In the dataset-level experiments, three out of the four baselines are sample-level MIA methods, and only one (DDI) is specifically designed for dataset-level MIA.

4. There is a lack of description regarding the implementation environment of the method (e.g., GPU specifications, runtime duration, and CPU resource usage).

**Questions:**

1. What is the relationship between LLM dataset-level MIA and the web?

2. Could more dataset-level MIA baselines be included in the comparison experiments to better validate the performance advantages of the proposed method?

3. How many queries are required to perform a dataset-level membership inference attack in a specific setting? When using the GPT-4 API, how many queries would be needed, and what would the associated cost be?

4. In practical scenarios, would the method’s high query volume potentially lead to detection by the target model, resulting in countermeasures?
5. Is this method a general-purpose approach for large models? Apart from LLMs and VLMs, could it be applied to other models such as those used in image recognition or audio processing?

**Reviewer Confidence:**

3: The reviewer is confident but not certain that the evaluation is correct

**Scope:**

2: The connection to the Web is incidental, e.g., use of Web data or API

---

### Official Review · Reviewer_XPzY · 2024-11-29

**Novelty:** 6
**Technical Quality:** 5

**Review:**

**Summary**

The paper introduces a dataset-level membership inference method based on Self-Comparison, offering an approach for detecting whether a dataset was used in the training of a target model. Instead of directly comparing member and non-member data, the method relies on paraphrasing the second half of sequences and evaluates likelihood changes pre- and post-paraphrasing. This approach does not require access to ground-truth member or non-member data from identical distributions. Instead, it uses only auxiliary non-member data from different distributions.

The method is evaluated against traditional Membership Inference Attacks (MIA) and dataset-level inference techniques, such as Distribution-based Dataset Inference (DDI) [1]. Testing spans diverse datasets and models, including publicly available models, fine-tuned models, and commercial API-based models. The evaluation metric, F1-score, is mainly used to assess the method’s ability to infer whether a dataset was part of a model's training set.

[1] Pratyush Maini, Hengrui Jia, Nicolas Papernot, and Adam Dziedzic. 2024. LLM Dataset Inference: Did you train on my dataset? arXiv preprint arXiv:2406.06443 (2024).
Strengths


**Major Strengths**

1. Comprehensive Evaluation - The evaluation is robust, spanning various datasets and models. It examines multiple metrics for confidence representation and includes comparisons against pre-trained models, fine-tuned models, and commercial API-based models.

2. Clear Intuition and Experimental Support - The methodology is well-explained with clear experimental backing. The paper highlights differences from current DDI approaches, emphasizing the distinct assumptions and the innovative use of half paraphrasing.

3. Coverage of Multiple Scenarios - The study addresses various scenarios, including attacks on Large Language Models (LLMs) and Vision-Language Models (VLMs). It also considers cases without access to model logits and partial dataset training conditions.

**Minor Strengths**

1. Clear Figure Presentation - The figures are well-designed, particularly the workflow visualization of the Self-Comparison framework. They effectively demonstrate the operations and logic behind half paraphrasing, significantly enhancing the reader’s understanding of the methodology.


**Major Weaknesses**

1. It is assumed that the model was trained on all dataset. You should evaluate the case where, for example, 30%, 50%, 60%, 80% of the dataset is used for training the model.

2. Limited Metric Usage - The study relies primarily on precision, recall, and F1-score. Incorporating additional domain-specific metrics would improve the evaluation’s depth.

3. Overemphasis in Writing - The writing style occasionally feels overly exaggerated and self-promotional, which could detract from the paper's objectivity.


**Minor Weaknesses**

1. Unclear Definition - The paper lacks clarity regarding the concept of protected data. More emphasis should be placed on defining the distinction between validation and protected data.

2. Parameter Selection Not Explained - The basis for choosing the parameters ϵ_1 and ϵ_2 is not explained. Expanding the experiment in Figure 7 to explore the attack results under different parameter values would strengthen the analysis.

2. Irrelevance of Table 2 - Table 2 seems unnecessary as it does not compare the proposed method to existing approaches. Moreover, the assumptions behind the attacks in the table are different, making it less relevant to the proposed methodology.

**Questions:**

1. Impact of Distribution - Does the auxiliary data’s distribution significantly affect the results? Would using auxiliary data from the same distribution (rather than differing distributions) yield different outcomes?

2. Thresholds for Attacks - Why not use the proposed threshold for every type of attack? This might simplify comparisons and ensure the fairness of experiments.

3. Do you assume that the model was trained only on the tested datasets? what if the model was trained on the tested dataset but on other datasets as well?

**Reviewer Confidence:**

4: The reviewer is certain that the evaluation is correct and very familiar with the relevant literature

**Scope:**

3: The work is somewhat relevant to the Web and to the track, and is of narrow interest to a sub-community

---

### Official Review · Reviewer_2XAc · 2024-12-04

**Novelty:** 5
**Technical Quality:** 6

**Review:**

The paper addresses challenges in dataset-level membership inference for large models like LLMs and VLMs. Existing methods for detecting whether specific datasets were used to train a model rely on strong assumptions, such as the availability of ground-truth member/non-member data with identical distributions. The authors propose Self-Comparison Membership Inference (SMI), a novel approach leveraging paraphrasing techniques to infer dataset membership, without requiring access to these strict assumptions. Numerical experiments demonstrate the superiority of their proposed method, although actual numbers seem to be slightly off (requires some explanation). Ablation studies and additional supporting evidence is sufficient and makes a compelling case for SMI's efficacy.

**Questions:**

1. In the introduction, the authors mentioned that the method of [29] is only applicable if there exists a validation dataset which has the same distribution as the protected data is available. Although the inadequacy of the method in [29] and the relative superiority of SMI is sufficiently explained, the methodology of DDI seems to be quite specialized and inherenty restrictive; does SMI hold overall advantages over other methods of determining membership in a dataset (which seems to be a well-researched problem)?
2. Numerical results show frequent occurances of high F1 scores, especially in Table 1. This raises the suspicion of overfitting, although it is unclear how overfitting should be understood within this work. Can the authors clarify on this point?

**Reviewer Confidence:**

3: The reviewer is confident but not certain that the evaluation is correct

**Scope:**

4: The work is relevant to the Web and to the track, and is of broad interest to the community